# DISENTANGLING TEXTUAL AND ACOUSTIC FEATURES OF SELF-SUPERVISED SPEECH REPRESENTATIONS

## ABSTRACT

Neural speech models build entangled internal representations, which capture a variety of features (e.g., pitch, loudness, syntax, or semantics of an utterance) in a distributed encoding. This complexity makes it difficult to track how such representations rely on textual and acoustic information when used in downstream applications, limiting their interpretability. In this paper, we build upon the Information Bottleneck principle to propose a disentanglement framework that separates speech representations learned by pre-trained neural speech models into two distinct components: one encoding content (i.e., what can be transcribed as text) and the other encoding acoustic features relevant to a downstream task. We apply and evaluate our framework to emotion recognition and speaker identification target tasks, quantifying the contribution of textual and acoustic features at each model layer. We also apply our disentanglement framework as an attribution method to identify the most salient speech frame representations from both the textual and acoustic perspectives.

## 1 INTRODUCTION

The internal activation vectors of most modern deep learning systems, including Neural Speech Models (NSM) such as Wav2Vec2 (Baevski et al., 2020), HuBERT (Hsu et al., 2021) are highly *entangled*. This means that distinct characteristics (such as pitch, loudness, syntax and semantics) of a spoken utterance are not separated into individual dimensions within the model's latent space, but are instead mixed together. Entanglement is a major obstacle for our ability to interpret and to intervene; *disentanglement*, to the extent that it is possible and even if imperfect, is therefore often highly desirable.

In this paper, we investigate to what extent we can learn to disentangle speech representations, focusing on the following desiderata:

- The method must disentangle representations already learned by large pre-trained spoken language models, with minimal extra data and computation.
- The method must be general, and applicable to representations from pre-trained models with varying architectures, sizes, and learning objectives.
- The approach should allow for acoustic features to emerge based on their relevance to a target downstream task, rather than being based on pre-specified static features such as pitch and timbre: if acoustic dimensions are defined in advance, some task-relevant aspects may be lost.

The above properties put strong constraints on our approach and distinguish our study from most work on learning disentangled representations for speech (Qian et al., 2019; 2020; Qu et al., 2024; Ju et al., 2024).

Although our proposed disentanglement framework is not modality-specific, in this paper we focus on two speech-related downstream tasks. As our first case study, we choose emotion recognition as the target task: while the content of an utterance is a strong cue for detecting emotion, the acoustic features provide additional information. For example, consider the utterances "I'm so happy" and "I'm fine,": where the former explicitly conveys happiness through its content, the latter can represent a wide range of emotions depending on prosody and tone. Our second case study closely follows the setup of the first, but we choose speaker identification as the target task. We hypothesize that content

offers only limited information, and the acoustic information will play an almost exclusive role in revealing the identity.

In Section 2, we propose a two-stage disentanglement framework (sketched in Figure1) based on the Information Bottleneck principle (Tishby et al., 2000; Alemi et al., 2016) to disentangle textual and acoustic features encoded in NSMs. In stage 1, we train a decoder with two objectives: to map the internal representation of an existing speech model to text, but also minimize the presence of irrelevant information in these representations. The goal is to ensure that the latent representation preserves only the speech features necessary for accurate transcription, while filtering out any extraneous characteristics. In stage 2, we train a second decoder on the same speech representations. This decoder also has access to the latent 'textual' representation learned in stage 1, and is again trained with 2 objectives: to predict our target labels (emotion or speaker IDs), and to minimize the amount of information encoded in the vector. This objective should ensure that the representation learned in stage 2 avoids encoding textual information – since the decoder already has access to it and the information minimization term discourages redundancy. Instead, it is expected to capture additional acoustic characteristics that are beneficial for the specific task.

In Section 3 we describe the models, dataset and training regime we use. In Sections 4 and 5 we evaluate our framework. We obtain highly compressed latent representations, that yield strong performance on both the standard transcription and our two target tasks. Moreover, our probing (Alain & Bengio, 2016) experiments show that the representations we obtain are almost perfectly disentangled from each other: while textual latent representations can predict transcriptions as effectively as the original speech representations, they exhibit random performance when predicting acoustic features (e.g., pitch or speaker identity). In contrast, acoustic representations excel at predicting these features but show random performance for transcriptions, validating the effectiveness of our disentanglement approach. The textual latent representations produced in the first stage are independent of the target task and can be easily applied to new downstream tasks.[1]

Finally, we analyze the emergence of the two types of representations in the original speech model. In Section 6, we trace back the contributions of the different layers of pre-trained and fine-tuned versions of the Wav2Vec2 (Baevski et al., 2020) to the representation of emotion. We find that as we progress through the layers, the acoustic contribution to emotion recognition significantly decreases in models fine-tuned on ASR, while the textual contribution increases, as they benefit from more accurate transcription and understanding of word polarity. Additionally, in Section 7, we qualitatively demonstrate that our disentanglement framework can serve as a feature attribution method to highlight the most significant frame representations for a given target task. Unlike existing gradient-based methods (Sundararajan et al., 2017), our approach allows us to determine whether a frame's contribution is textual or acoustic. Such disentangled attribution techniques can have many applications, e.g., in psychiatric research where the mismatch between textual and acoustic emotion expression was shown predictive of mood and disorders (Niu et al., 2023) or in bias control within speech agents.

## 2 DISENTANGLEMENT FRAMEWORK

This section explains how we build upon the Information Bottleneck principle to disentangle textual and acoustic information within speech representations that contribute to a targeted downstream task.

### 2.1 VARIATIONAL INFORMATION BOTTLENECK

The core idea of an Information Bottleneck (IB, Tishby et al., 2000) is to learn a stochastic encoding $Z$ that maximizes the prediction of a target variable $Y$, while minimizing the information retained about the input $H$. Accordingly, the loss function to be optimized based on this principle can be defined as follows:

$$\mathcal{L}_{\text{IB}} = I(Z, Y) - \beta I(H, Z) \tag{1}$$

where $I(., .)$ represents mutual information that measures the dependence between two variables. The coefficient $\beta \geq 0$ controls the trade-off between retaining information about either H or Y in Z.

---

[1]We will publicly release the code and the disentangled models.

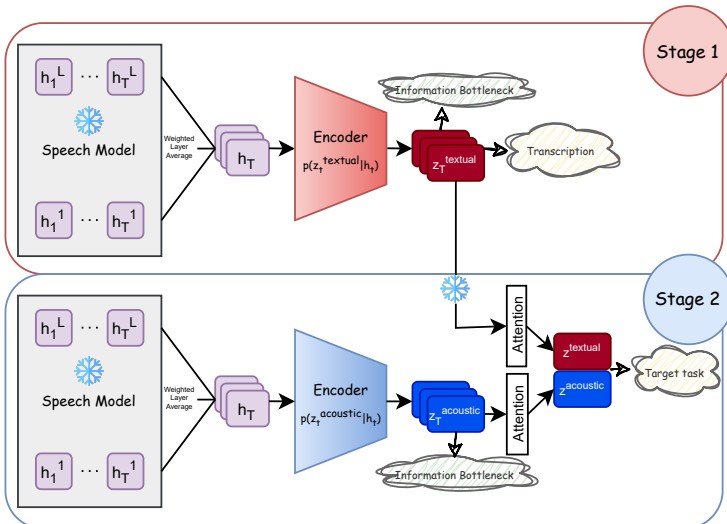

Figure 1: An overview of our disentanglement framework: In stage 1, hidden states derived from a frozen speech model are compressed using VIB to decode transcription. In stage 2, the hidden states are compressed to decode emotion, conditioned on the frozen textual latent representations learned in stage 1. This procedure results in disentangled latent representations for textual and acoustic features relevant to the target task within each speech frame representation.

The exact computation of IB loss, however, is intractable. To address this, a Variational Information Bottleneck (VIB, Alemi et al., 2016) has been proposed, it establishes a lower bound on the IB loss in Equation 1. The VIB loss is defined as[2]

$$\mathcal{L}_{\text{VIB}} = \underbrace{\frac{1}{N}\sum_{n=1}^{N}\mathbb{E}_{z\sim p_\theta(z|h_n)}[\log p_\phi(y_n|z)]}_{\text{Task loss}} - \beta\underbrace{\frac{1}{N}\sum_{n=1}^{N}\text{KL}[p_\theta(z|h_n), r(z)]}_{\text{Information loss}} \qquad (2)$$

where $N$ denotes the sample size, $p_\phi(y|z)$ acts as the decoder, a neural network that predicts the label $y$ from the latent representation $z$,[3] while $p_\theta(z|h)$ serves as a stochastic encoder, mapping the input $h$ to the representation $z$; $r(z)$ approximates the marginal $p(z)$. The first term in the loss function encourages the encoder to preserve information relevant to the label, while the second term, the KL divergence, pushes it to discard as much information as possible. The approximate marginal $r(z)$ is typically assumed to be a spherical Gaussian (Alemi et al., 2016). The encoder $p(z|h)$ is parameterized using an MLP to predict the mean $\mu$ and the diagonal covariance matrix $\Sigma$: $p_\theta(z|h) = \mathcal{N}(z|\mu_\theta(h), \Sigma_\theta(h))$. The optimization is performed using the reparameterization trick (Kingma & Welling, 2013).

## 2.2 OUR PROPOSED METHOD

Consider a sequence of speech representations $(\boldsymbol{h}_1, ..., \boldsymbol{h}_T)$, produced by a pre-trained speech model, where $\boldsymbol{h}_t$ represents the speech frame representation at timestamp $t$, and $T$ denotes the total number of frames in a given utterance. Given a target task, our goal is to decompose each frame representation into two distinct latent representations: $\boldsymbol{z}_t^{\text{textual}}$ and $\boldsymbol{z}_t^{\text{acoustic}}$. Our approach to disentangling these two latent representations involves two stages, sketched in Figure 1. Both stages use frozen speech representations obtained from the same speech model as input.

---

[2]See Alemi et al. (2016) for derivation.

[3]Formally, this represents a variational approximation of the true $p(y_n|z)$, which is intractable as VIB assumes that the joint distribution factorizes as $p(y_n, z, h_n) = p(h_n)p(z|h_n)p(y_n|h_n)$ (Alemi et al., 2016).

### 2.2.1 STAGE 1

In the first stage, we aim to distill the textual content from speech frame representations, while discarding other non-textual features. The textual capability of a speech model is typically evaluated using automatic speech recognition (ASR), and measured by the word error rate (WER) metric, which assesses how accurately the model can transcribe spoken utterances based on their representations.

We extract all speech frame representations for a given audio from a pre-trained speech model and compute a weighted average across model layers, with the weights learned during training. These frame representations are then given as input to a bottleneck encoder, which compresses the information into low-dimensional latent frame representations. To decode transcriptions from the latent frame representations, we employ the Connectionist Temporal Classification (CTC, Graves et al., 2006) loss as the task loss term in Equation 2, thus minimizing the following loss function:

$$\mathcal{L}_{\text{CTC}} - \beta \mathcal{L}_{\text{Information}} \tag{3}$$

In this way, we force the bottleneck encoder $p(\boldsymbol{z}^{\text{textual}}|\boldsymbol{h})$ to retain only the information necessary for transcription (as encouraged by the CTC loss), while discarding irrelevant features in the original representation ($\boldsymbol{h}$) (constrained by the information loss[4]). We refer to the latent frame representation for frame $t$ learned at this stage as $\boldsymbol{z}_t^{\text{textual}}$. Intuitively, these latent representations capture the minimum statistics of speech representations needed for decoding transcriptions.

### 2.2.2 STAGE 2

Our goal in the second stage is to capture acoustic features that complement the textual features learned in stage 1 and contribute to the downstream task. To achieve this, we replace the task loss in Equation 2 with the Cross-Entropy loss (CE) over the target class labels, thus, minimizing the following loss function:

$$\mathcal{L}_{\text{CE}} - \beta \mathcal{L}_{\text{Information}} \tag{4}$$

Using labeled data for our target task in this stage, we extract speech frame representations from the same speech model and again learn a weighted average over layers. These representations are then fed into a bottleneck encoder $p(\boldsymbol{z}^{\text{acoustic}}|\boldsymbol{h})$ — with the same architecture as the encoder in stage 1 — to form the complimentary latent representations ($\boldsymbol{z}_t^{\text{acoustic}}$). We apply the information loss to the frame-wise latent representations at the output of the encoder. Subsequently, we pass these latent representations through an attention layer[5] to have latent representations at the utterance level, since labels for our target tasks are assigned to the whole utterance. Finally, the pooled latent representation $\boldsymbol{z}^{\text{acoustic}}$ is concatenated with the frozen textual latent representations $\boldsymbol{z}^{\text{textual}}$ (previously trained in Stage 1) to decode the target task. This conditional setup encourages the trainable latent representations to retain only non-textual features, particularly those absent in the textual latent representations. During the training process in this stage, no gradient updates are directed back to the $\boldsymbol{z}_t^{\text{textual}}$ learned in stage 1.

## 3 EXPERIMENTAL SETUP

### 3.1 TRAINING DETAILS

For training VIB, we follow Alemi et al. (2016) and model $r(\boldsymbol{z})$ and $p(\boldsymbol{z}|\boldsymbol{h})$ as multivariate Gaussian distributions: $r(\boldsymbol{z}) = \mathcal{N}(\boldsymbol{z}|\boldsymbol{\mu} = \mathbf{0}, \boldsymbol{\Sigma} = \mathbf{1})$ and $p(\boldsymbol{z}|\boldsymbol{h}) = \mathcal{N}(\boldsymbol{z}|\boldsymbol{\mu}(\boldsymbol{h}), \boldsymbol{\Sigma}(\boldsymbol{h}))$. The bottleneck encoders to estimate $\boldsymbol{\mu}$ and $\boldsymbol{\Sigma}$ consist of two shared linear layers with the same dimensionality as the original hidden representations ($\boldsymbol{h}_t$), followed by independent $d$-dimensional layers for each, with GELU (Hendrycks & Gimpel, 2016) activation functions in between. We experiment with different bottleneck dimensions $d = \{16, 32, 64, 128, 256\}$ as the output size of the bottleneck encoders. To estimate the gradients, we employ the reparameterization trick (Kingma & Welling,

---

[4]In an ablation experiment (Appendix H), we also trained the encoders without the information loss. Although this led to a slight improvement in performance, it failed to disentangle the textual and acoustic information as intended (shown in Section 5), which confirms the central role of the information loss term in our framework.

[5]We choose an attention layer over a simple average pooling to later recognize the key frame representations for the sake of interpretability.

2013): $z_t = \mu(h_t) + \Sigma(h_t) \odot \epsilon$, where $\epsilon$ is sampled from the normal distribution $\mathcal{N}(\mathbf{0}, \mathbf{1})$. During inference, we use $z_t = \mu(h_t)$. In our implementation, we gradually increase the $\beta$ coefficient linearly from 0.1 to 1 during training. For decoding, we utilize a randomly initialized linear projection into $C$ classes, where $C = 32$ for transcription (corresponding to the number of target characters), $C = 4$ for emotion recognition, and $C = 24$ for speaker identification. The training hyperparameters are detailed in Appendix A.

## 3.2 TARGET MODELS

To obtain speech representations, we use two prominent self-supervised speech models: Wav2Vec2 (Baevski et al., 2020) and HuBERT (Hsu et al., 2021) in their different sizes: Base (12-Transformer layers, 768-hidden size) and Large (24, 1024), obtained from the HuggingFace (Wolf et al., 2020) library. Our experiments include both pre-trained (on raw speech) and fine-tuned[6] (on transcribed speech) versions of these models. Both models employ the Transformer (Vaswani et al., 2017) architecture and learn speech representations through masked prediction in a self-supervised manner. Wav2Vec2 employs a contrastive loss to identify the masked speech frame among distractors, while HuBERT uses k-means clustering to create prediction targets. The models are further fine-tuned with additional labeled speech data by optimizing a linear classifier using CTC loss to decode transcription.

## 3.3 DATA

**Transcription.** For training in stage 1, we use subsets of two widely used read speech corpora: LibriSpeech (Panayotov et al., 2015) and Mozilla's Common Voice 17.0 (Ardila et al., 2019). The former is derived from audiobooks, while the latter is recorded by contributors reading sentences displayed on a screen. We randomly select 4,000 examples from each corpus, ensuring an equal representation of gender and speaker ID, with each sample having a maximum duration of 14 seconds. This results in 17.4 hours of transcribed speech training data. To evaluate transcription performance, we use the entire test-clean set of the LibriSpeech dataset. Following (Baevski et al., 2020), we remove non-spoken special characters (e.g., commas and periods) from transcriptions, as these are not included in our target vocabulary.

**Target tasks.** For emotion data in stage 2, we utilize IEMOCAP (Busso et al., 2008) database, which consists of five dyadic sessions involving ten speakers (5 male, 5 female). Following prior research (Li et al., 2021; 2022), we exclude utterances without transcripts and combine *Happy* and *Excited* labels to form a 4-way classification task. We then undersample the dataset to balance emotion classes, resulting in 4,064 utterances ($\sim 5$ hours of audio, with an average duration of 4.4 seconds per segment). Each utterance is assigned one emotion from the label set: {*Angry*, *Happy*, *Neutral*, *Sad*}. For Speaker Identity as our target task, we utilize a subset of Mozilla's Common Voice 17.0 dataset (Ardila et al., 2019), consisting of 4,000 training and 1,000 test samples, stratified by two genders (male and female) and 24 speaker identities. All audio files in this study are resampled to 16 kHz to match the sampling rate used for the pre-training data of the target models.

## 4 TASK PERFORMANCE AFTER VIB TRAINING

In this section, we test the effectiveness of the disentangled representations against the original entangled representations on the downstream tasks. We will verify the disentanglement in the next section. For comparison, we also train identically structured decoders which rely on the original hidden states ($h_t$); the difference with VIB training is that the information loss is excluded during the training process. The performance of these classifiers (which can be viewed as a repurposed form of *probing* (Alain & Bengio, 2016; Tenney et al., 2019)) serves as a strong baseline, representing the performance achievable relying on the hidden states, without compressing them into latent representations.

Table B.1 reports both VIB and probing performances, averaged over three runs with different random seeds, for transcription, emotion recognition, and speaker identification tasks. For the transcription task at stage 1, VIB demonstrates a similar or sometimes even lower word error rate (WER) compared

---

[6]The HuBERT model lacks a released fine-tuned checkpoint in the Base size.

| Model | Size | Stage 1 Transcription (WER ↓) | | Stage 2 Emotion (Acc. ↑) | | Stage 2 Speaker Id. (Acc. ↑) | |
|---|---|---|---|---|---|---|---|
| | | Probing | VIB | Probing | VIB | Probing | VIB |
| HuBERT | Base | 45.7 | 41.6 | 61.8 | 62.7 | 97.8 | 99.1 |
| | Large | 36.0 | 37.6 | 57.3 | 66.1 | 92.3 | 98.4 |
| Wav2Vec2 | Base | 50.1 | 45.0 | 61.4 | 58.9 | 99.8 | 97.9 |
| | Large | 46.9 | 48.7 | 62.2 | 65.9 | 97.9 | 99.8 |
| HuBERT-FT | Large | 3.1 | 20.8 | 54.7 | 64.8 | 90.5 | 98.2 |
| Wav2Vec2-FT | Base | 5.7 | 12.7 | 63.5 | 56.6 | 99.6 | 98.2 |
| | Large | 3.5 | 8.3 | 62.1 | 63.4 | 98.5 | 99.6 |

Table 1: VIB ($d = 128$) and probing performance. Lower WER and higher Accuracy are better. Random baselines: WER $= 100$ for transcription, Accuracy $= 25$ for emotion, and Accuracy $= 4.1$ for speaker identification. Scores are averaged over three runs with different random seeds.

to probing classifiers, implying the success of VIB training in compressing essential information for audio transcription (WER $= 100$ serving as the performance of the random baseline).[7] As expected, representations derived from fine-tuned models show better transcription performance compared to those from pre-trained models (for both VIB and probing) as they are specifically tuned for transcription. The same success is evident in decoding our target tasks in stage 2, where we report the accuracy of both VIB and probing classifiers for emotion recognition and speaker identification tasks (the random baseline accuracy is 25 and 4.1, respectively). Overall, the classifier benefits more from the specialized, compressed representations than from the original hidden states, as VIB encourages retaining only task-relevant information, resulting in more robust representations.

We also found VIB performance consistent across various bottleneck dimensions; see the results in Appendix D. Both VIB and probing learn weights for layer averaging (see Figure 1). These weights are necessary because the information is not uniformly distributed across the NSM layers. As shown in Appendix E, both approaches use these weights; the weights provide insights into the contribution of individual layers to the acoustic and textual features.

## 5 EVALUATION OF DISENTANGLEMENT

### 5.1 SANITY CHECK PROBING

The product of our training procedure (described in Sec. 2) are disentangled latent representations $z_t^{\text{textual}}$ and $z_t^{\text{acoustic}}$ for each speech frame representation $h_t$. In this section, we investigate these latent representations to validate if they are truly disentangled by probing them for various types of textual and acoustic information. Given an audio input, we expect that $z_t^{\text{textual}}$, which is specialized for text, should not encode any aspects of the acoustic characteristics of the audio. In contrast, $z_t^{\text{acoustic}}$, which is specialized for audio features, should not contain any information about the audio transcription.

### 5.1.1 SETUP

To assess textual capability, we train a linear probing classifier on frozen latent representations to predict their transcriptions. To estimate acoustic capability, we train a set of probing classifiers followed by an attention layer on latent representations to predict various acoustic features at the utterance level, including Mean Intensity, Mean Pitch, Gender, and Speaker Identity. For comparison, we also probe the original hidden states for the same objectives with the identically-structured classifiers. We utilize a subset of Mozilla's Common Voice 17.0 dataset (Ardila et al., 2019), consisting of 4,000 training and 1,000 test samples, stratified by two genders (male and female)[8] and 24 speakers. We extract labels for acoustic features directly from the raw audio waveforms using the

---

[7]We also evaluated the transcription performance using character error rate (CER) as an additional metric; results are in Appendix C.1.

[8]The exclusion of other groups is due to the binary labeling in the dataset, rather than a choice by the authors.

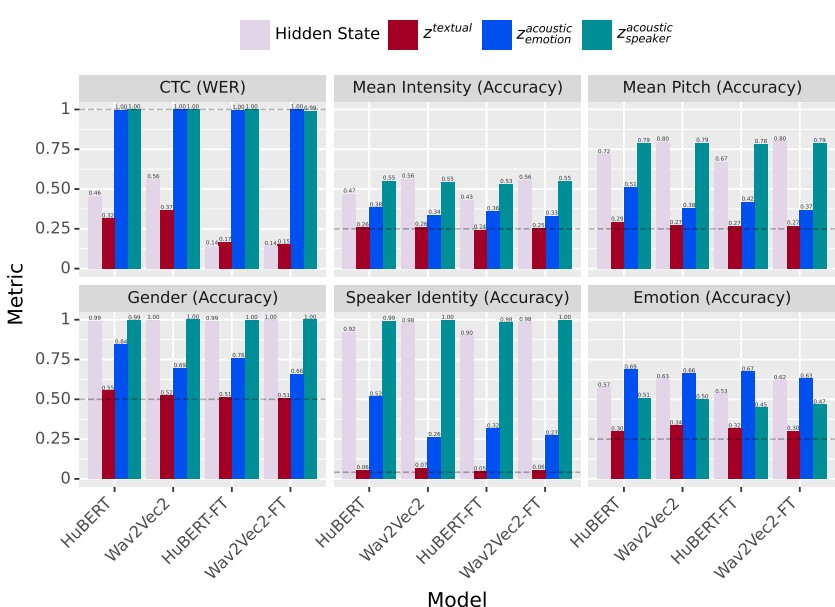

Figure 2: Probing performances of latent representations ($d = 128$) learned at stages 1 and 2, along with hidden states derived from Large models for transcription and a set of audio features. For the WER metric, the lower score is better, while for other metrics, the higher is better. The dashed line in each plot represents the random baseline.

Parselmouth toolkit (Jadoul et al., 2018). We then discretize them into four equally sized buckets based on quantiles to cast the task as a four-way classification problem.

### 5.1.2 RESULTS

Figure 2 illustrates the classification results for Large models. Dashed lines represent the random baseline performance. As we can see, acoustic latent representations (for both target tasks; $z_{\text{emotion}}^{\text{acoustic}}$ and $z_{\text{speaker}}^{\text{acoustic}}$) exhibit no awareness of the textual content of the audio as their performance for CTC matches the random baseline (WER $= 1$). Conversely, textual latent representations are as effective as the original hidden states and – for pre-trained models – even outperform them at decoding transcription.

Looking into predicting acoustic features, textual latent representations consistently show random performance, suggesting no acoustic features are encoded within them. Acoustic latent representations, however, show substantial probing performance despite not having any explicit acoustic objective in their training at stage 2. Interestingly, acoustic latent representations for the task of speaker identification are better at encoding acoustic features than those of emotion recognition. It is likely due to acoustic information playing a greater role in revealing the speaker identity.

Additionally, in contrast to $z_{\text{speaker}}^{\text{acoustic}}$, acoustic latent representations for emotion ($z_{\text{emotion}}^{\text{acoustic}}$) do not match the performance of the original hidden states. For example, for the Wav2Vec2 model, the probing performance for Speaker ID based on hidden states is 0.98, while it is 0.26 for acoustic latent representations (the random baseline is $1/24 \approx 0.04$). This disparity suggests that not all those acoustic features encoded in the hidden representations are crucial for recognizing emotion, thus, not all of those features were retained in stage 2. These findings could be important in real-world scenarios where, for privacy protection, encoding acoustic features without precisely identifying the speaker can be essential. The pattern of the probing results is also similar for Base models and for various bottleneck dimensions, as reported in Appendix G.

### 5.2 QUALITATIVE EVALUATION

Next, we visualize the latent representations to gain insight into how they have been encoded in the representation space for emotion recognition. We make use of RAVDESS (Livingstone & Russo,

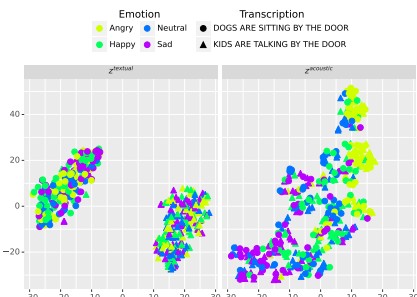

Figure 3: t-SNE plots of the textual and acoustic latent representations for Wav2Vec2-Large, marked and colored according to their transcription and emotion labels, respectively.

2018) dataset, which contains only two identical statements spoken by 24 actors (12 male, 12 female) with 4 different emotions. This makes it ideal for this analysis, as the linguistic content remains constant across utterances with different emotions.

We select examples with matching emotion labels from the IEMOCAP dataset, resulting in 384 utterances (each averaging 3.7 seconds). We obtain the speech representations from the large Wav2Vec2 model and generate their corresponding latent representations using the bottleneck encoders trained in stages 1 and 2 by doing only inference without any further training. Then, we compute the average of frame representations over all frames in each utterance and apply t-SNE.

Figure 3 shows a 2D projection of these latent representations with data points marked and colored according to their transcription and emotion labels, respectively. The textual latent representations (learned in stage 1) are perfectly clustered according to their transcriptions. In contrast, the acoustic latent representations (learned in stage 2) are not separated by transcription, suggesting that no textual information is retained there. Instead, these acoustic representations are roughly clustered by emotion, which nicely aligns with our desired goal. The pattern is also the same for the Base model, reported in Appendix I.1.

## 6 LAYERWISE EMOTION CONTRIBUTION

With our disentangled framework established and validated, we can now quantify, separately, the extent to which each layer in a speech model contributes textually and acoustically to the target task of emotion recognition. We first train the latent representations using the same setup as we described in Section 2, but instead of using a weighted layer average in the input, we use the frame representations from a specific layer of the model. The layerwise results for stages 1 and 2 are shown in Figure 4, with the black horizontal dashed line representing random performances.

For HuBERT, the transcription ability improves as we move through the layers. In contrast, Wav2Vec2 shows a U-shaped pattern, with the best performance in the middle layers and the final layers being unable to decode transcription. For Wav2Vec2-FT, however, transcription performance improves sharply in the last layers, which is expected given that the model is fine-tuned for ASR, indicating substantial changes in the last layers during fine-tuning. In stage 2 (middle panel), models perform best at emotion recognition in the middle layers. Compared to pre-trained, the fine-tuned model (Wav2Vec2-FT) shows a significant decline in emotion recognition performance in the final layers, where these layers lose acoustic information in favor of encoding text transcription.

### 6.1 LAYERWISE EMOTION PROBING

We next train a probing classifier (followed by an attention layer) on top of frozen $z_t^{\texttt{textual}}$ and $z_t^{\texttt{acoustic}}$ latent representations to decode emotion. Each latent representation is specialized to preserve either textual or acoustic features from the original hidden states of each model layer. Therefore, the probing performance reveals the extent to which these textual and acoustic features contribute to emotion recognition. For comparison, we also train the same probing classifier on the original hidden states across each layer.

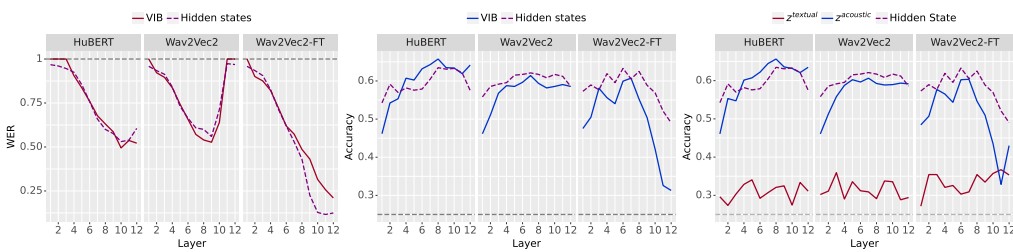

Figure 4: Layerwise performance of VIB ($d\!=\!128$) for transcription at stage 1 (*left*) and emotion classification at stage 2 (*middle*), compared to layerwise performance using original hidden state activations. *Right*: layerwise textual and acoustic contribution to emotion classification, compared to layerwise performance using hidden states of Base models. Lower WER and higher accuracy are better. The black horizontal dashed line represents the random baseline.

Figure 4 (right) shows emotion probing performance. Comparing pre-trained and fine-tuned models, the latent acoustic representations learned from the final layers of Wav2Vec-FT contribute significantly less to emotion recognition. This is likely due to the model losing some acoustic information during fine-tuning in favor of transcription capabilities. However these layers benefit more from textual features in predicting emotion, as their representations offer more accurate transcriptions. Compared to Wav2Vec2, HuBERT shows a greater contribution to emotion recognition; acoustically in the middle layers, and textually in the last layer.

## 7 LOCALIZING SALIENT TEXTUAL AND ACOUSTIC FRAMES

In this section, we use our disentanglement framework to localize salient input features for the target tasks. The attention layer in stage 2 of our framework (see Figure 1) can be used to identify those frames in the original audio input whose latent representations contribute most to our target tasks. Crucially, the disentanglement mechanism allows us to clearly separate the contributions of acoustic features from those of textual features. This disentangled attribution could be particularly useful in fields like psychiatry, where differences between textual and acoustic emotional expressions can aid in the diagnosis of disorders (Niu et al., 2023), or in detecting bias in speech agents' responses to user requests.

Finding salient input features for model decisions is often done by computing the gradient of the model's output with respect to the inputs (Sundararajan et al., 2017; Ancona et al., 2018; Yuan et al., 2019; Samek et al., 2019). To compare our attention scores with gradient-based attribution scores, we train another classifier for the target task on the original hidden states of the Wav2Vec2 model and use Integrated Gradients (IG) to compute attribution to individual frames (Sundararajan et al., 2017). We normalize the IG scores to sum to 1.

We then obtain frame-by-frame distributions of both acoustic and sentiment features to compare them directly with attribution results. For acoustic features, we focus on intensity and pitch, identifying and representing peaks and valleys in their temporal patterns. To analyze sentiment, we use the spaCy toolkit (Honnibal & Montani, 2017) to annotate word-level polarity (positive, negative, or neutral) within the utterances. Using Montreal Forced Aligner (McAuliffe et al., 2017), we map these word-level labels to frames (see details in Appendix J). As a result, each feature vector for a speech frame includes the following dimensions: (1) the presence of a peak or valley in intensity, (2) the same for pitch, and (3) the presence of a 'sentiment-laden' word, all normalized to the range $[0, 1]$.

We then compute the dot product between the frame-wise attribution scores and the corresponding feature vectors. Figure 5 presents these results, averaged across all examples in the IEMOCAP test set, and demonstrates that acoustic attention effectively captures peaks and valleys in acoustic features (intensity and pitch), while textual attention focuses on word polarity. Both have higher agreement with features than Integrated Gradient scores (see Appendix K for qualitative examples). Note that textual attention exhibits fairly high similarity with acoustic features as polar words are often pronounced with emotional emphasis.

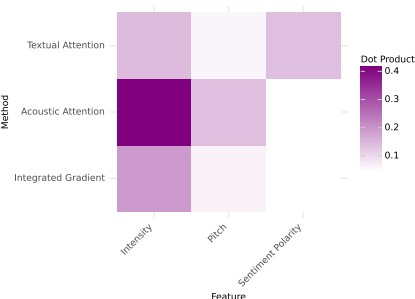

Figure 5: Dot product of attribution scores and different acoustic and textual cues.

## 8 RELATED WORK

Learning disentangled representations such that distinct factors are separable and controllable has been the focus of many studies. Bengio et al. (2013) motivate this approach within the context of deep learning, while Wang et al. (2024) provide a general up-to-date overview. For speech, disentanglement has been used for controllable style transfer in voice conversion, where the goal is to modify para-linguistic information while preserving linguistic content (van den Oord et al., 2017; Polyak et al., 2021; Qian et al., 2019; 2020; Choi et al., 2021; 2022; Qu et al., 2024). In our study, we specifically posit the following desiderata which distinguish our study from most work on learning disentangled representations for speech:

- The approach disentangles representations learned by pre-trained models, with minimal data or computation. This contrasts with methods (such as AutoVC (Qian et al., 2019), SpeechSplit (Qian et al., 2020), Prosody2Vec (Qu et al., 2024), FACodec (Ju et al., 2024)) that rely on learning disentangled representations from scratch, or require large amounts of fine-tuning data: training of Prosody2Vec takes around three weeks. In contrast, we train our proposed method directly on the target task, and it takes only a few minutes.

- The approach applies to pre-trained models with different architectures. This contrasts with methods tailored to a specific architecture: for example, Prosody2Vec (Qu et al., 2024) builds on top of HuBERT's (Hsu et al., 2021) quantization module to encode content.

- The approach should allow for acoustic features to emerge based on a target task. In contrast, an approach such as SpeechSplit (Qian et al., 2020) splits audio into pre-specified factors (content, timbre, pitch, rhythm).

From the perspective of employing VIB to maximize the mutual information between latent representations and target labels (as opposed to using Autoencoders and reconstruction loss), our work relates to (Gao et al., 2021; Pan et al., 2020) in computer vision. Regarding feature attribution, our work relates to (Wang et al., 2023), which proposes a multi-modal information bottleneck approach aimed at disentangling relevant visual and textual features to enhance the interpretability of vision-language models.

## 9 CONCLUSIONS

We present a disentanglement framework which separates entangled representations of NSMs into textual and acoustic components, while retaining only features relevant to the target tasks. In experiments with the emotion recognition and speaker identification tasks, we demonstrated that the framework can effectively isolate key features, improving interpretability, whilst maintaining the performance of the original model. The framework holds potential for applications where disentanglement is key, such as ASR systems where privacy-preservation is key for responsible deployment (and sometimes even a legal requirement); separating sensitive speaker-specific information from task-relevant information is essential in such systems. We also showed that the the framework is useful for disentangled feature attribution, revealing the most significant speech frames from both textual and acoustic perspectives.

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

## A  TRAINING HYPERPARAMETERS

In our experiments, all models were trained for 50 epochs using the AdamW optimizer with gradient norm clipping. The learning rate was set to $1e{-}3$ and $1e{-}4$ for Base and Large models, respectively, with a warmup ratio of $0.1$ and a cosine decay scheduler, together with weight decay. Specifically for training transcription prediction using CTC loss, we use a batch size of $1$; the effective batch size here is the number of frame representations since the CTC loss is computed across all frames. For other training objectives, we use a batch size of $8$.

| Model | Size | LibriSpeech WER/CER ($\downarrow$) | | Common Voice WER/CER ($\downarrow$) | |
|---|---|---|---|---|---|
| | | Probing | VIB | Probing | VIB |
| HuBERT | Base | 45.7 / 12.7 | 41.6 / 10.8 | 65.3 / 25.0 | 64.0 / 23.9 |
| | Large | 36.0 / 9.3 | 37.6 / 8.9 | 53.5 / 18.6 | 55.1 / 19.2 |
| Wav2Vec2 | Base | 50.1 / 13.9 | 45.0 / 11.6 | 68.1 / 26.1 | 67.8 / 26.1 |
| | Large | 46.9 / 12.2 | 48.7 / 12.0 | 63.9 / 22.4 | 61.0 / 21.4 |
| HuBERT-FT | Large | 3.1 / 0.7 | 20.8 / 6.9 | 21.4 / 8.4 | 39.7 / 17.9 |
| Wav2Vec2-FT | Base | 5.7 / 1.2 | 12.7 / 2.3 | 30.7 / 12.7 | 34.9 / 13.8 |
| | Large | 3.5 / 0.8 | 8.3 / 1.6 | 25.6 / 10.2 | 27.3 / 10.6 |

Table B.1: VIB ($d = 128$) and probing performance at stage 1 on the entire test sets of two different speech corpora: LibriSpeech and Mozilla's Common Voice 17.0. Lower WER and CER are better. Random baselines: WER/CER $= 100$. Scores are averaged over three runs with different random seeds.

# B  TRANSCRIPTION EVALUATION FOR MORE DATASETS

# C  TRANSCRIPTION EVALUATION WITH OTHER METRICS

# D  ABLATION ON THE BOTTLENECK DIMENSION

# E  LAYER WEIGHT AVERAGING

In the input of both VIB and probing training setups in Section 4, we train weights for layer averaging (see Figure 1). Let us have a look at these trained layer weights to see the contribution of each layer at each stage. Figure E.1 shows these layer weights for both Base and Large models. Overall, layer weights learned in the VIB setup closely follow the weights trained with original hidden states in the probing setup.

In stage 1 we observe that, in pre-trained models, the upper-middle layers are more capable of decoding transcription. In fine-tuned models, however, this information shifts to and becomes concentrated at the final layer. This pattern is consistent across both Base and Large model sizes as well as for different bottleneck dimensions.

For emotion recognition and speaker identity tasks in stage 2, however, there is a notable difference between Base and Large models. In Base models, the earlier layers contribute more in both pre-trained and fine-tuned setups. In contrast, layers in Large models contribute uniformly, suggesting that the acoustic information useful for these target tasks is distributed across layers in Large models.

# F  SANITY CHECK PROBING EXPERIMENT FOR OTHER EXISTING METHODS

In this section, we compare the disentanglement effectiveness of our proposed method with FACodec (Ju et al., 2024) and SpeechSplit (Qian et al., 2020), two existing methods mentioned by one of the reviewers. We extracted the embeddings of the disentangled components from their framework and trained a linear classifier to probe them for various types of textual and acoustic information (similar to the probe in Section 5.1).

Figures F.1 and F.2 demonstrate the results for FACodec and SpeechSplit, respectively. We can see that their acoustic components perform poorly in linearly decoding acoustic features compared to our acoustic latent representation (Figure 2). Moreover, it is not possible to linearly decode the transcription (content) of the audio from their content vectors (in contrast to ours).

This suggests that while existing work on speech disentanglement has mainly focused on reconstructing input audio with different styles or speakers, the disentanglement quality of their components has not been properly evaluated. For instance, in Section 4 in (Ju et al., 2024), it is stated: *"For Word Error Rate (WER), we use an ASR model (footnote: HuBERT-Large fine-tuned on the 960 hours of*

*LibriSpeech for ASR) to transcribe our generated speech."* However, this approach is not an accurate way of assessing the quality of content vectors since the HuBERT model here acts as a confounding variable that significantly contributes to the reported WER. In contrast, our CTC probing method directly evaluates the ease of decoding transcriptions from these content vectors, providing a more reliable measure of their disentanglement quality.

## G   REPLICATION OF SANITY CHECK PROBING EXPERIMENT FOR OTHER MODEL SIZES AND BOTTLENECK DIMENSIONS

## H   SANITY CHECK PROBING EXPERIMENT FOR THE SETUP WITHOUT THE INFORMATION LOSS CONSTRAINT

## I   REPLICATION OF T-SNE VISUALIZATION

## J   ALIGNING FRAME REPRESENTATIONS WITH AUDIO FRAMES AND TRANSCRIPTION

To map the frame representations to their original input frames and their corresponding word transcriptions, we use Montreal Forced Aligner (McAuliffe et al., 2017, MFA). Using this tool, we extract the start time ($t_s$) and the end time ($t_e$) of each word in an utterance, and map them to boundary frames $f_s$ and $f_e$:

$$f = \lceil \frac{t}{\mathcal{T}} \times T \rceil \tag{5}$$

where $\mathcal{T}$ and $T$ denote the total time of a given audio and the total number of frames in the frame representation, respectively.

## K   QUALITATIVE EXAMPLES FOR FEATURE ATTRIBUTION

Figure K.1 showcases the textual and acoustic attention scores derived from our disentanglement framework for Wav2Vec2, together with the gradient scores and framewise acoustic features (pitch and intensity), for two utterances from the IEMOCAP (Busso et al., 2008) test dataset labeled '*Happy*'. Both figures are vertically segmented according to the word's time stamps in the utterances. In the left panel, the textual attention highlights part of the frames corresponding to the positive words "THANKS" and "NICE", while, the acoustic attention peaks at a frame where there is a drastic change for both pitch and intensity, at the beginning of the pronunciation of the word "OH,". In the right panel for the second utterance, we can see a uniform pattern for textual attention, which makes sense as there is no textual cue in the utterance for emotion recognition. However, the acoustic attention again shows significant peaks where pitch and intensity change drastically. In contrast, Integrated Gradient tends to highlight the silent parts of the audio.

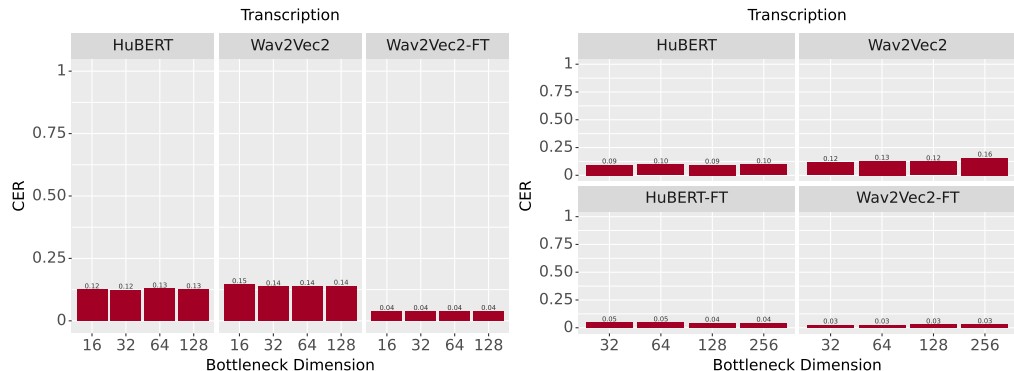

Figure C.1: VIB performance at stage 1, evaluated by Character Error Rate (CER) metric, across various bottleneck dimensions for **Base** (left) and **Large** (right) models. CER is reported for a random subset consisting of 3.2 hours from a combination of the LibriSpeech and Mozilla Common Voice 17.0 datasets.

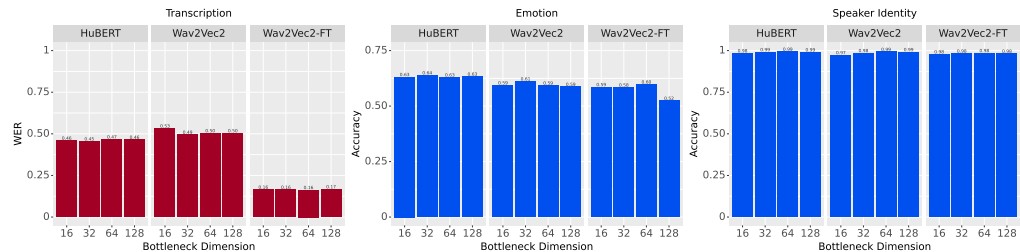

Figure D.1: VIB performance across various bottleneck dimensions for **Base** models. WER is reported for a random subset consisting of 3.2 hours from a combination of the LibriSpeech and Mozilla Common Voice 17.0 datasets.

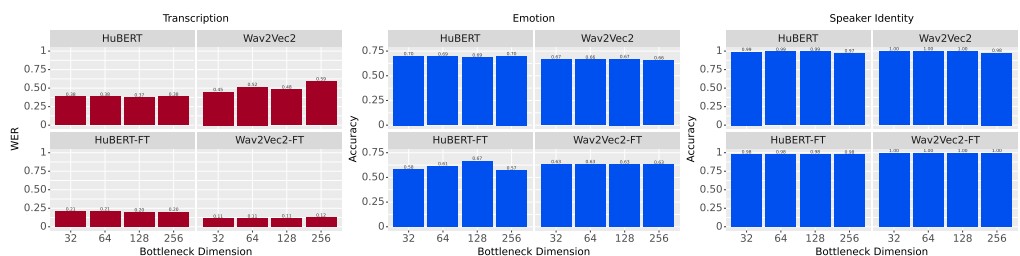

Figure D.2: VIB performance across various bottleneck dimensions for **Large** models. WER is reported for a random subset consisting of 3.2 hours from a combination of the LibriSpeech and Mozilla Common Voice 17.0 datasets.

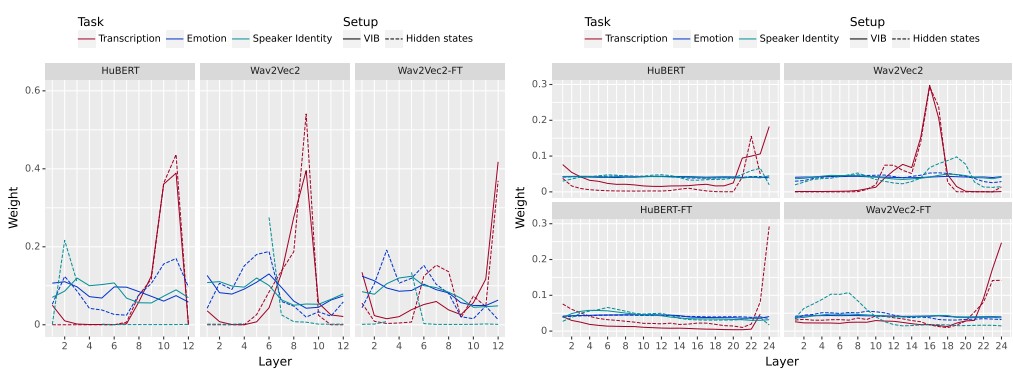

Figure E.1: Layer weights of **Base** (left) and **Large** (right) models, trained at each VIB stage, compared to the probing on hidden states.

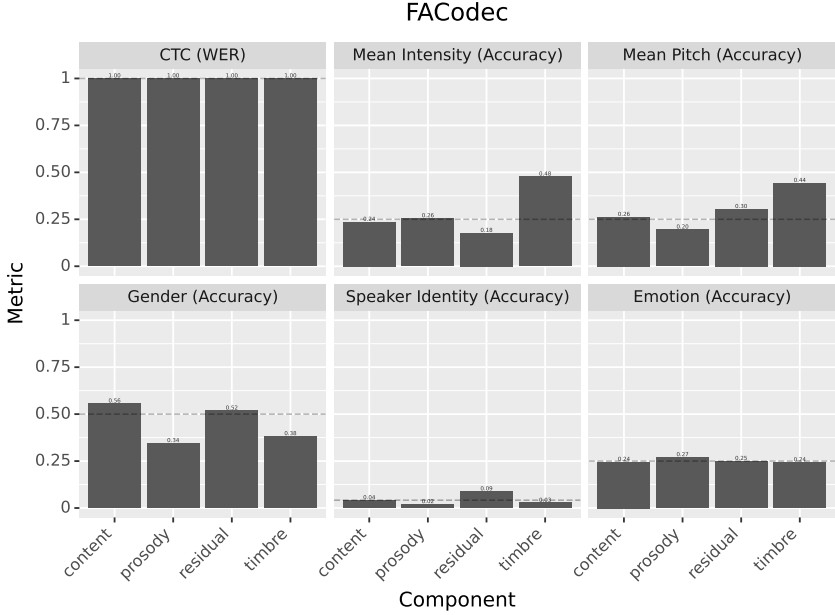

Figure F.1: Probing performances of disentangled representations from FACodec framework for transcription and a set of audio features

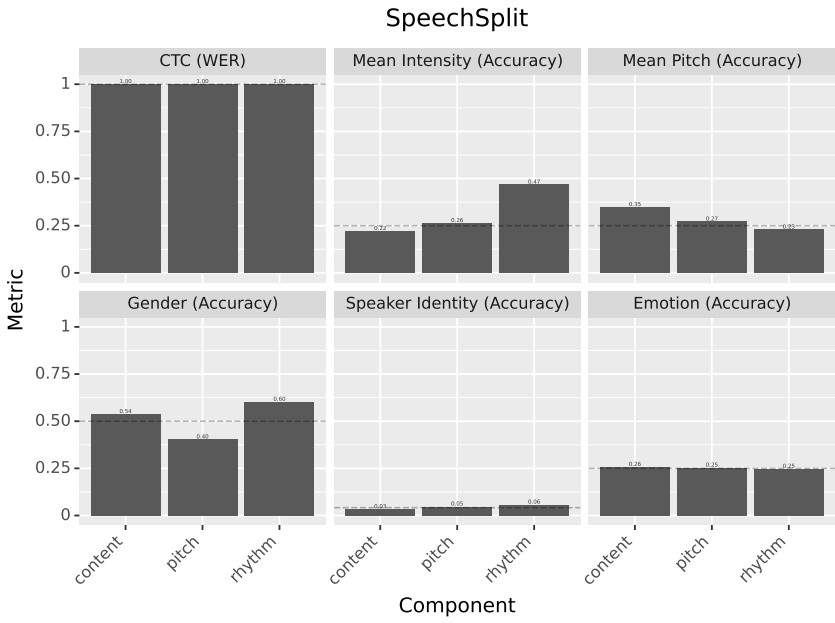

Figure F.2: Probing performances of disentangled representations from SpeechSplit framework for transcription and a set of audio features

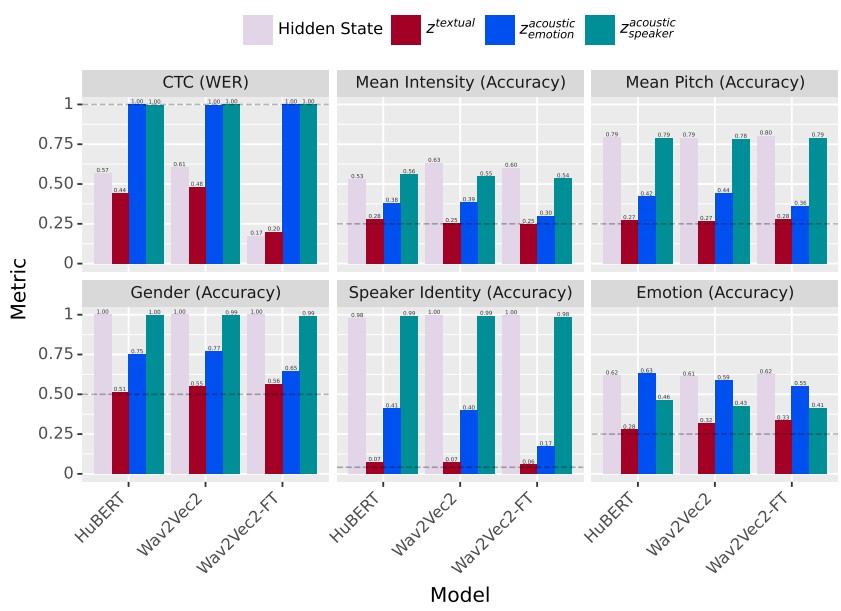

Figure G.1: Probing performances of latent representations ($d=128$) learned at stages 1 and 2, along with hidden states derived from **Base** models for transcription and a set of audio features.

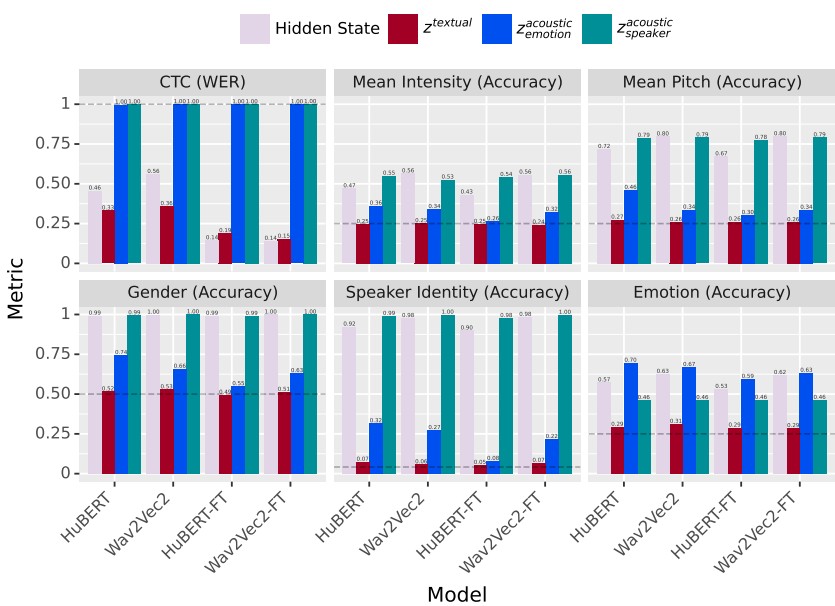

Figure G.2: Probing performances of latent representations ($d=32$) learned at stages 1 and 2, along with hidden states derived from **Large** models for transcription and a set of audio features

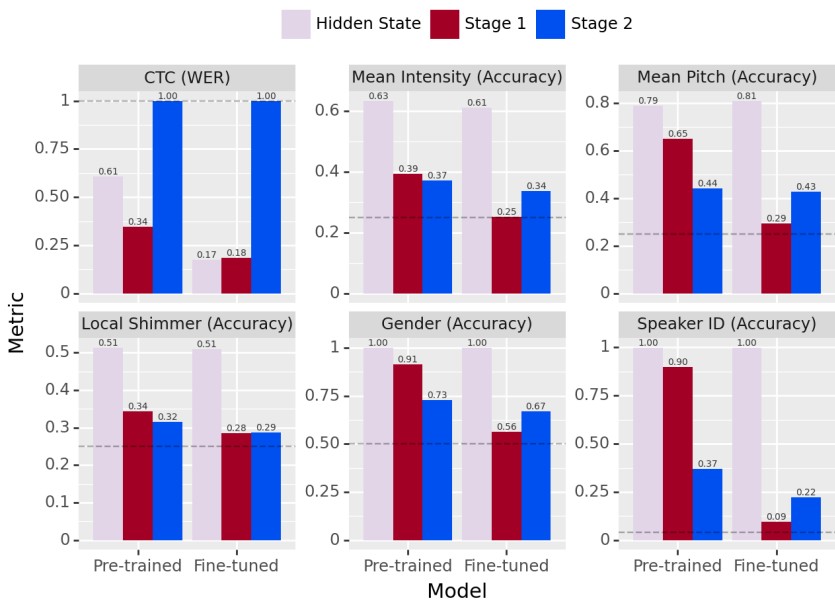

Figure H.1: Probing performances of latent representations learned at stages 1 and 2 **without information loss constraint**, along with hidden states derived from **Wav2Vec2-Base** model for transcription and a set of audio features. This clearly shows that excluding the information loss in the training fails to disentangle textual and acoustic information.

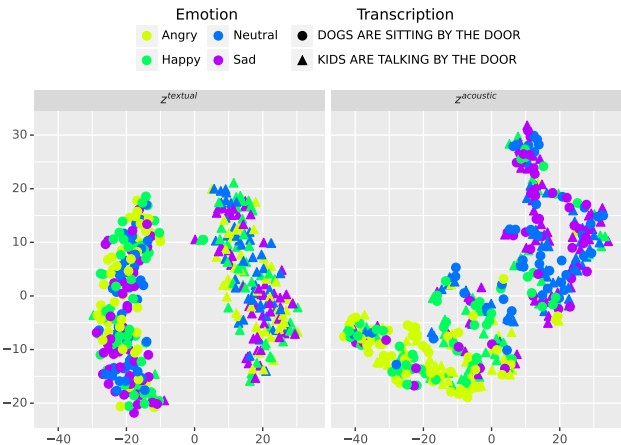

Figure I.1: t-SNE plots of the textual and acoustic latent representations for the Wav2Vec2-Base model, marked and colored according to their transcription and emotion labels, respectively.

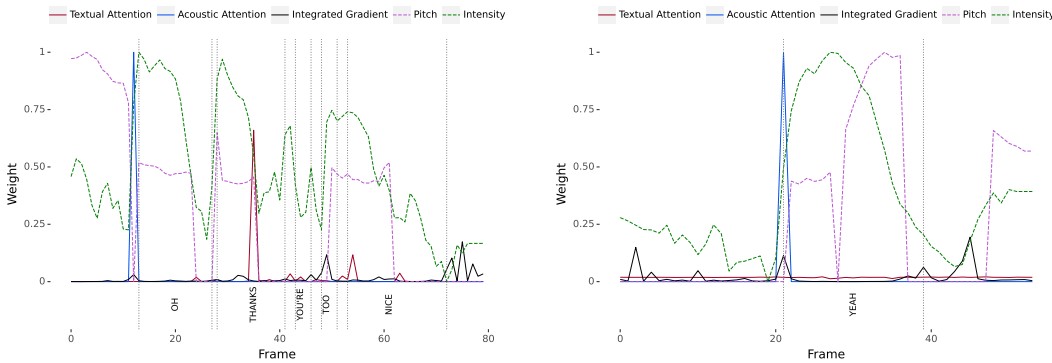

Figure K.1: Textual and acoustic attentions, and Integrated Gradient scores, together with two framewise acoustic features (Pitch and Intensity) for two utterances labeled *Happy*.

