# OpenReview forum: "Disentangling Textual and Acoustic Features of Neural Speech Representations"
_ICLR.cc/2025/Conference — Submitted to ICLR 2025_

### Official Review · Reviewer_icVr · 2024-10-27

**Soundness:** 3
**Presentation:** 3
**Contribution:** 2
**Rating:** 3
**Confidence:** 4

**Summary:**

This paper proposes a new framework for disentangling speech representations from neural speech models (like Wav2Vec2 and HuBERT) into two distinct components: textual content (what can be transcribed as text) and acoustic features (like emotion or speaker identity). This separation is important because neural speech models typically create deeply entangled internal representations that combine various features, making it difficult to isolate specific information or suppress potentially sensitive acoustic features (such as gender or speaker identity) in real-world applications.
The authors present a two-stage training framework based on the Variational Information Bottleneck technique. In the first stage, a decoder is trained to map speech representations to text while minimizing irrelevant information from input, ensuring only features necessary for transcription are preserved. In the second stage, another decoder is trained that has access to the textual representation from previous stage and is trained to predict target labels for downstream task while minimizing information encoding. They evaluated their framework on emotion recognition and speaker identification tasks, demonstrating that the resulting representations were effectively disentangled - the textual representations could predict transcriptions but performed randomly when predicting acoustic features, while acoustic representations showed the opposite pattern.
The authors also analyzed how different layers of pre-trained and fine-tuned Wav2Vec2 models contribute to emotion recognition. They found that in models fine-tuned for automatic speech recognition (ASR), the acoustic contribution to emotion recognition decreases in higher layers while the textual contribution increases. Additionally, they showed that their framework can serve as a feature attribution method to identify the most significant frame representations for a given task, distinguishing between textual and acoustic contributions.

**Strengths:**

The main strengths of the paper are as follows:
1. The authors provide a clear motivation and explanation for the problem under consideration.
2. The method is clearly explained, creating no confusion in grasping the idea.
3. The experiment section is well-written with relevant experiments
4. The authors answer some key questions related to the work such as extent of disentanglement and its benefits
5. The last section of the paper talks about prior works which are in the same domain to provide readers an idea about the novelty in this work.
6. The authors have further cited rsome extremely elevant works.

**Weaknesses:**

Here are the main weaknesses:
1. I struggle to understand the new idea in this work because the VIB technique has existed for a while.
2. The concept of employing neural networks to learn or estimate bounds on Mutual information has existed for a long time (see
    a. MINE: Mutual Information Neural Estimation by Mohamed Ishmael Belghazi, Aristide Baratin, Sai Rajeswar, Sherjil Ozair, Yoshua
        Bengio, Aaron Courville, R Devon Hjelm.
    b. DEEP VARIATIONAL INFORMATION BOTTLENECK by Alexander A. Alemi, Ian Fischer, Joshua V. Dillon, Kevin Murphy
    c.  Representation Learning with Contrastive Predictive Coding by Aaron van den Oord, Yazhe Li, Oriol Vinyals
3. The authors do not provide explanation in Table 1 regarding why WER increase for Fine-tuned models after disentanglement training.
4. In Figure 2, there seems to be some strange behavior as far as prosody prediction is concerned. Pitch, intensity, rhythm, voice quality, etc have been identified as key contributors to the perception of emotion from speech. It makes little sense as to why the disentangled acoustic representation would remove that information.
5. It has been shown before that different layers of Self-supervised models (HuBERT and W2V2) learn different types of representation from speech signal (acoustic, prosody and semantic). Therefore, section 6 reaffirms those prior studies while providing no new information for the infromed readers.

**Questions:**

NA

---

### Official Review · Reviewer_s8oL · 2024-10-31

**Soundness:** 2
**Presentation:** 3
**Contribution:** 3
**Rating:** 6
**Confidence:** 4

**Summary:**

The paper uses the Variational Information Bottleneck framework to separate textual and acoustic features of representations from SSL speech models, such as HuBERT and wav2vec2. This approach involves two stages: first, it isolates textual information by training models to transcribe content with minimized other unrelated information. The second stage targets acoustic features for tasks like emotion and speaker recognition.
They validate the proposed method through experiments on ASR, emotion recognition and speaker identification, showing its effectiveness in distinguishing between acoustic and textual attributes. This approach also has potential applications in privacy preservation, where disentangling speaker identity from transcription could help secure ASR systems.

**Strengths:**

1. The paper is well-written and is easy to follow.
2. The proposed approach is easy to use.
3. Experiments in Section 6 align in part with the findings of previous work on layer-wise speech SSL models[1], reflecting the effectiveness of the proposed method.


[1] A. Pasad, B. Shi and K. Livescu, "Comparative Layer-Wise Analysis of Self-Supervised Speech Models," ICASSP 2023 - 2023 IEEE International Conference on Acoustics, Speech and Signal Processing (ICASSP),

**Weaknesses:**

1. There is previous work using the Information Bottleneck for feature disentanglement, such as in [2] and [3]. It would be better to cite these studies and highlight the distinctions between this paper and prior work.
2. Experiments comparing the proposed method with existing approaches are lacking. As there are lots of works for speech representation disentanglement like AutoVC, SpeechSplit, or FAcodec[4] , it would strengthen the paper to report the performance of at least one existing methods.
3. In Table 1, VIB loses essential information for textual representation, resulting in a much higher WER compared to Probing for HuBERT-FT and Wav2Vec2-FT. Training on a different dataset with positive outcome might help alleviate this issue.


[2] Pan, Z., Niu, L., Zhang, J., & Zhang, L. (2021). “Disentangled Information Bottleneck.” *Proceedings of the AAAI Conference on Artificial Intelligence*

[3] Gege Gao, Huaibo Huang, Chaoyou Fu, Zhaoyang Li, Ran He; “Information Bottleneck Disentanglement for Identity Swapping." Proceedings of the IEEE/CVF Conference on Computer Vision and Pattern Recognition (CVPR), 2021, pp. 3404-3413

[4] Ju, Zeqian, et al. "Naturalspeech 3: Zero-shot speech synthesis with factorized codec and diffusion models." arXiv preprint arXiv:2403.03100 (2024).

**Questions:**

1. What is the reason for using the mixture of LibriSpeech and Common Voice?
2. The WER reported in Table 1. seems to be higher than expected. What is the possible reason for that? For LibriSpeech, what subset is used in the experiments? LibriSpeech-clean or LibriSpeech-other?
3. The Information Bottleneck (IB) method focuses on retaining only information that’s relevant for predicting the target variable, filtering out anything unnecessary. This makes it dataset-dependent. For instance, when I train the stage 2 framework on a dataset for emotion recognition, the disentangled features capture emotional information but lacks speaker-specific information. I wonder if it would be possible to handle both speaker recognition and emotion recognition in stage 2, so that we preserve both emotion-related and speaker-related information. Alternatively, we could consider adding a stage 3 focused on speaker identity, while stage 2 remains dedicated to emotion recognition.

---

> ### Author Response · Authors · 2024-11-25
> **Response to reviewer s8oL (1)**
>
> Many thanks for your detailed review!
>
> __[Experiments comparing the proposed method with existing approaches are lacking. As there are lots of works for speech representation disentanglement like AutoVC, SpeechSplit, or FAcodec, it would strengthen the paper to report the performance of at least one existing method]__
>
> Thank you for raising this point – we regret not having been more explicit about this in the original submission. The key point to make here is that, while it is true that there is lots of work for speech representation disentanglement, our work is not directly comparable to them. In our study, we specifically posit the following desiderata:
> - The approach must be able to disentangle representations already learned by large pre-trained models of spoken language, with minimal extra data and limited computational resources. We thus do not consider methods that rely on learning disentangled representations from scratch, or require large amounts of fine-tuning data.
>     - For example, AutoVC, SpeechSplit, Prosody2Vec, and FACodec, all have their own complex modeling architecture (including a deep stack of CNNs and LSTMs), and trains on raw audio to reconstruct the input audio signals, and this is why they are evaluated for style transfer in voice conversion application.
>     - For example, as stated by the authors in the paper, the entire pretraining procedure for Prosody2Vec takes around three weeks, as their parameters are first pre-trained on spontaneous and emotional speech and then fine-tuned for the target task (emotion). In contrast, we train our proposed method directly on the target task, and it takes only a few minutes.
> - The approach must be general and applicable to representations learned by pre-trained models with different architectures, sizes, and learning objectives. We thus do not consider methods that are tailored to a specific architecture.
>     - e.g., Prosody2Vec which builds on top of HuBERT's quantization module to encode content.
> - The approach should allow for acoustic features to emerge based on their relevance to a target task (such as emotion or speaker identification), rather than being shaped around pre-specified static features such as pitch and timbre. This last point is important because defining acoustic dimensions in advance might result in losing some task-relevant aspects of acoustic representations.
>     - For example, existing work pre-specify the following components regardless of the target task:
>         - AutoVC: content, style
>         - SpeechSplit: content, timbre, pitch, rhythm
>         - Prosody2Vec: content, speaker, prosody
>         - FACodec: content, prosody, timbre, and residual acoustic
>
> The above desired properties put strong constraints on our approach and distinguish our study from all the existing work on learning disentangled representations for speech that we are aware of. We do, however, present a serious effort to compare to FACodec and SpeechSplit in our revised manuscript (Appendix F);

---

> > ### Author Response · Authors · 2024-11-25
> > **Response to reviewer s8oL (2)**
> >
> > __Q1: [What is the reason for using the mixture of LibriSpeech and Common Voice?]__
> >
> > LibriSpeech consists of audiobooks, while CommonVoice includes recordings from the general public. In combination, they create a more comprehensive dataset for transcription, leading to improved performance and, hence, resulting in better task-independent textual features. Although using each dataset alone might result in a slight degradation in WER, in our ablation experiments, we observed a similar pattern in the disentanglement evaluation, nullifying the sensitivity of the source of transcribed data used in stage 1 in our disentanglement framework.
> >
> > __Q2: [The WER reported in Table 1. seems to be higher than expected. What is the possible reason for that?]__
> >
> > Pre-trained models do not have very strong transcriptional information (the final layers do not have any sense of transcription, and most of the information comes from the middle layers). During the common ASR fine-tuning process, these last layers undergo significant changes. Thus, learning a latent representation that is robust to noise and discards irrelevant information makes pre-trained representations more resilient and less sensitive to noise, allowing them to perform better than entangled representations.
> >
> > __Q2: [For LibriSpeech, what subset is used in the experiments? LibriSpeech-clean or LibriSpeech-other?]__
> >
> > Subsets of LibriSpeech-train-clean and LibriSpeech-test-clean were used for training and test, respectively.
> >
> > __Q3: [The Information Bottleneck (IB) method focuses on retaining only information that’s relevant for predicting the target variable, filtering out anything unnecessary. This makes it dataset-dependent. For instance, when I train the stage 2 framework on a dataset for emotion recognition, the disentangled features capture emotional information but lacks speaker-specific information. I wonder if it would be possible to handle both speaker recognition and emotion recognition in stage 2, so that we preserve both emotion-related and speaker-related information. Alternatively, we could consider adding a stage 3 focused on speaker identity, while stage 2 remains dedicated to emotion recognition]__
> >
> > This is an interesting alternative setup, but the quick way to compute task-dependent latent representations is actually there by design, as different types of tasks may require different aspects of acoustic features. This is one of the key factors that differentiates our work from approaches that attempt to disentangle speech representations into a pre-specified set of elements using reconstruction loss (VAE). We believe our work fills a gap in the existing literature, by providing a general framework for disentangling textual from acoustic aspects of speech representations. Ours allows for acoustic features to emerge based on their relevance to a target task, rather than being shaped around pre-specified features such as pitch, and timbre. This is important because defining pre-specified acoustic features in advance might result in losing some task-relevant aspects of acoustic representations.

---

> ### Comment · Reviewer_s8oL · 2024-11-27
> **Official Comment by Reviewer s8oL**
>
> Thank you for your response.
>
> The rebuttal addresses most of my concerns and highlights an additional strength I previously overlooked: the efficiency of training in terms of dataset size and time consumption. The revisions strengthen the paper.
>
> However, I still have a major concern:
> > Q2: [The WER reported in Table 1. seems to be higher than expected. What is the possible reason for that?]
> >
> > Pre-trained models do not have very strong transcriptional information (the final layers do not have any sense of transcription, and most of the information comes from the middle layers). During the common ASR fine-tuning process, these last layers undergo significant changes. Thus, learning a latent representation that is robust to noise and discards irrelevant information makes pre-trained representations more resilient and less sensitive to noise, allowing them to perform better than entangled representations.
>
> Specifically, my concern relates to the WER of VIB when applied to HuBERT-FT and Wav2Vec2-FT. I think they should achieve much lower WERs on LibriSpeech and Common Voice datasets (as observed in Probing tasks).
>
> For the reasons mentioned above, I will raise my score to marginally above the acceptance threshold, as I believe the motivation and the efficiency of the proposed method are strong. However, it appears to somewhat compromise textual content information.

---

> > ### Author Response · Authors · 2024-11-29
> >
> > __[I still have a major concern: Specifically, my concern relates to the WER of VIB when applied to HuBERT-FT and Wav2Vec2-FT. I think they should achieve much lower WERs on LibriSpeech and Common Voice datasets (as observed in Probing tasks).]__
> >
> > Thanks again for your constructive criticism! We believe we can address your remaining concern.  The issue stems from the fact that, in these experiments, the transcription classifier in the _'VIB'_ setup is trained during stage 1, where it learns to transcribe from stochastic states influenced by noise. This, coupled with joint optimization (as the vector z textual evolves during training), makes optimization more challenging and results in suboptimal performance when applied to non-noisy final z textual (i.e., its means: $z_t = \mu(h_t)$) as shown in Table 1. In contrast, the classifier in the _'Probing'_ setup is trained directly on the original (entangled) representations, which are unaffected by noise and static when training the classifier.
> >
> > In contrast, if we train a new, randomly initialized classifier on the final, trained textual z representations – obtained without applying any noise (i.e. means) – we achieve the same WER as when using the original entangled representations. This is exactly how we evaluated our disentanglement in Sec. 5. This is shown in the panel labeled _‘CTC’_ in Figure 2, where we train a new classifier on original and learned textual z representations to decode transcription and achieve the same results in both entangled and disentangled setups for both HuBERT-FT and Wav2Vec2-FT. (Figures G.1 and G.2, moreover, show that this is consistent for other model sizes and bottleneck dimensions).
> >
> > Importantly, the final classifier at stage 1 was only for training purposes (and for analysis in Table 1) and in practice we never use it again: in Figure 1, we throw out the final classifiers and take the textual (red) and acoustic (blue) encoders as the results of our training procedure. In other, the real evaluation of how good our disentangled textual component is what we did in (Section 5, Figure 2).
> >
> > We will address this in the next version of the manuscript, as the paper cannot be revised at this time.

---

### Official Review · Reviewer_8vUa · 2024-11-01

**Soundness:** 3
**Presentation:** 2
**Contribution:** 2
**Rating:** 3
**Confidence:** 4

**Summary:**

This paper describes an application of information bottleneck training to isolate aspects of a speech representation.  The description of the approach is quite clear.  The paper includes a variety of analyses of the learned representations that show that disentangling is achieved.

**Strengths:**

The described approach is sensible and its specifics are clearly described.

There are a number of interesting analyses based on probing experiments to attempt to identify what information is still available in different layers of the network and assessment of the information related to distinct tasks in different frames of the input audio.

**Weaknesses:**

The motivations in the abstract and conclusion are not well connected to the modeling and analysis.  E.g. one motivating application is to minimize the privacy risk from encoder representations.  This hasn't been assessed in the model or paper.

The disentangling approach is based on supervised tasks.  The contributions necessary for emotion classification or speaker id.  It is unclear how these learned representations would transfer to some new task. Would this approach need to be extended to a "stage 3 training process?

Multiple training stages incur additional complexity.  It would be interesting to see if these multiple objectives would be included into a single stage training.

The impact on performance in Table 1 does not deliver a consistent message.  The Transcription show substantial regressions in both of the FT representations.  The improvements to Emotion and Speaker Id are stronger but more consistent on the large sized models while on the Base sizes, there are regressions on the wave2vec variants.  This sensitivity to SSL objective and model size suggests that this approach may not be robust to new tasks or architectures.

**Questions:**

Why subsample Librispeech and Common Voice so heavily for the transcription task?  Librispeech contains 960h of transcribed audio, but this approach uses less than 20.

How important is the ordering of the tasks?  Would the performance be identical if Emotion or Speaker Id were stage 1 and Transcription was stage 2?

---

> ### Author Response · Authors · 2024-11-25
> **Response to reviewer 8vUa**
>
> Many thanks for your detailed review!
>
> __[The motivations in the abstract and conclusion are not well connected to the modeling and analysis. E.g. one motivating application is to minimize the privacy risk from encoder representations. This hasn't been assessed in the model or paper]__
>
> Thank you for this feedback. We clarified the motivation in the introduction and removed the reference to the privacy risks.
>
> __[It is unclear how these learned representations would transfer to some new task. Would this approach need to be extended to a "stage 3 training process?]__
>
> There is no need for a stage 3. The textual latent representations produced in stage 1 are independent of the target task and can be applied to new downstream tasks. As a result, for each new task, one can directly start with stage 2 to train the shallow encoder and classifier on the target task. We believe this is, in fact, a strong point of our approach, as it reduces the required compute considerably when there are many such new tasks.
>
> __[Multiple training stages incur additional complexity. It would be interesting to see if these multiple objectives would be included into a single stage training]__
>
> Joint training is certainly feasible. However, training in stages offers key advantages: Stage 1 can be trained once and reused across all tasks, while Stage 2 training is highly computationally efficient. Moreover, we are publicly releasing the Stage 1 models, enabling follow-up work to easily build on our method by focusing solely on Stage 2 training. From a paper perspective, this approach simplifies the discussion of results, ensuring comparability and robustness across tasks since they all share a common starting point in Stage 1.
>
>
> __[Inconsistent task performance improvement/degradation for different objectives and model sizes]__
>
> First and foremost, we emphasize that our paper primarily focuses on disentanglement. These experiments were conducted to demonstrate that the disentanglement method does not result in any significant trade-off in performance in downstream task performance (Emotion and Speaker Id). For transcription (which is not the focus of this work), increase in WER from using VIB is expected for models fine-tuned on large datasets. This is because stage 1 was conducted on smaller datasets, which can inadvertently suppress some of the WER capabilities. Apart from this, identifying a clear pattern may be challenging. We could speculate that there is a trade-off at play: disentangled representations can facilitate the extraction of task-specific information, thereby enhancing accuracy, but the information bottleneck can also to the loss of potentially beneficial information, which could negatively affect accuracy. This trade-off likely varies depending on the model's architecture and size. While the difference in performance varies, the results collectively suggest that probing has a largely neutral impact on model performance while offering the added benefits of interpretability and control.
>
> __Q1: [Why subsample Librispeech and Common Voice so heavily for the transcription task? Librispeech contains 960h of transcribed audio, but this approach uses less than 20]__
>
> In our initial experiments, we found that a few hours of transcribed data were actually sufficient to train a classifier for decoding transcription. According to Tables 9 and 10 of Appendix C in Wav2Vec2 [1], using even only 1 hour of transcribed data is enough to achieve ASR performance comparable to fine-tuning on the full 960 hours. Please note that in contrast to ASR fine-tuning, the model's weights in our approach are fixed; we only train an encoder bottleneck together with a linear classifier. While increasing the scale of transcribed data in stage 1 might improve CTC performance, it would not harm the core observation here, as we compare the performance using the entangled representations (probing) with the disentangled ones (VIB), both of which utilize the same transcribed data.
>
> [1]: https://arxiv.org/pdf/2006.11477
>
> __Q2: [How important is the ordering of the tasks? Would the performance be identical if Emotion or Speaker Id were stage 1 and Transcription was stage 2?]__
>
> Indeed, it matters! For example, when it comes to emotion, both the content of the audio (text) and the acoustic features can help recognize emotion. Our goal here is to measure the contribution of each of the textual and acoustic features to the target task. To address this, we first separate the textual features at stage 1, and then focus on finding the remaining features at stage 2, which go beyond text but actually contribute to the target task. Training the emotion at stage 1 would contaminate acoustic features with textual ones since textual features are also relevant to the target task.

---

> > ### Author Response · Authors · 2024-11-29
> >
> > We would like to thank you again for your detailed and thoughtful feedback. We believe we have addressed the concerns you and the other reviewers raised, and we kindly ask that you review the improvements and clarifications we have made.
> > Specifically, we have:
> > - Clarified our motivation and the Targeted Setting / Desiderata.
> > - Strengthened the experimental evaluation by incorporating comparison to previous methods (FACodec and SpeechSplit) in Section F in the appendix (Figures F.1 and F.2).
> > - Thoroughly revised the Related Work discussion to provide a clearer explanation of how our approach aligns with, diverges from, and contributes to the existing body of research on disentanglement in speech processing.
> >
> > Please let us know if our responses properly address your concerns.

---

### Official Review · Reviewer_HyNY · 2024-11-03

**Soundness:** 2
**Presentation:** 3
**Contribution:** 2
**Rating:** 3
**Confidence:** 3

**Summary:**

Many standard speech representations are learned in a self-supervised way (HuBERT, w2v2, etc) and hence are, essentially, entangled blackboxes that have acoustic and textual features mixed in them in an arbitrary way. One can imagine scenarios where this is undesired, and it would be better to have a control over what/how features are encoded. This paper proposes a method building such disentangled representations, using the IB principle. As a running example, the paper opposes information that encodes textual content and acoustic information, encoding emotion or speaker identity. The paper shows that their method successfully disentangles the inputs features (variants of HuBERT, W2V2). The authors conclude with several interpretability studies of the models that use those features.

**Strengths:**

* I find the "disentanglement evaluation" part pretty convincing.

**Weaknesses:**

* The proposed method assumes that we have labeled tasks for all potential downstream tasks. So one starts with general-purpose self-supervised representations such as HuBERT -- which are entangled -- and ends up with representations that are (a) disentangled, but (b) are likely only useful for the tasks where we have labels. In this scenario, I am not entirely convinced that one has to use VIB (see the questions below).

* I am not entirely convinced by the motivations of the paper. If one needs to be confident that models do not use non-textual information while taking decisions, they can train models to make those decisions using pure transcripts. This is a simple baseline solution the paper should be having in mind.

* There are some concerns wrt the experimental setup -- see Questions 2, 3, 4.

**Questions:**

1. L30 mentions that Whisper has highly entangled representations, in the same list as HuBERT or Wav2Vec2. It is never mentioned/evaluated later; is there any evidence that it is likely to have as entangled representations as SSL models from this list? It is trained purely in a supervised way for text transcription/translation, iirc, hence I would assume it learns purely text-focused features.

2. Do I understand it correctly that non-standard splits of LibriSpeech are used for the purpose of "ensuring an equal representation of gender and speaker ID" (S3.3) Is there a strong reason for that in the text transcription tasks? For the sake of comparability with all the existing literature, I would advise using standard some dev/test-{clean,other} splits.

3. Having a single linear probing classifier gets WER of ~50 for W2V2 and HuBERT. Only the pre-finetuned models get reasonable error rates. Is this a good evaluation setup to draw conclusions from?

4. What dataset is used to calculate WER in Table 1? Is this a mix of LibriSpeech and CommonVoice? Those are very different datasets, it would make sense to report them separately.

5. At least a part of motivation of the work is that by using disentangled representations one can be confident that model is using the features it is allowed to. For instance, the model doesn't rely on leaking gender or voice information when making text-based decisions. I generally get the idea, but the transcription vs gender/emotion classification task split is not a particularly convincing combination. If we are worried that the model uses something beyond the text content when making some downstream decisions, we can replace it with an (ASR + text classifier) model. Can we think of a more convincing scenario?

6. Do we actually need VIB? How different it would be if we used the labels to train a combination of ASR, Speaker and Emotion classifiers and used their outputs?

---

> ### Author Response · Authors · 2024-11-25
> **Response to reviewer HyNY (1)**
>
> Many thanks for your detailed review!
>
> __[The proposed method assumes that we have labeled tasks for all potential downstream tasks. So one starts with general-purpose self-supervised representations...and ends up with representations that are (a) disentangled, but (b) are likely only useful for the tasks where we have labels. In this scenario, I am not entirely convinced that one has to use VIB]__
>
> We focus on scenarios where a speech model is fine-tuned for a downstream task, a widely used approach to adapt general-purpose models to specific applications. Our method leverages the same target task labels while incorporating an information bottleneck loss to achieve disentangled representations. Through our experiments (see Section 5), we show that using the Variational Information Bottleneck (VIB) effectively ensures that acoustic information is excluded from the textual representation and textual information is absent from the acoustic component.
>
> __Q6: [Do we actually need VIB? How different it would be if we used the labels to train a combination of ASR, Speaker and Emotion classifiers and used their outputs?]__
>
> We replicated our training procedure for Wav2Vec2-base to train the encoders without the information loss constraint and reported the results in Appendix (Figure H.1). Although this led to a slight improvement in task performance, it failed to disentangle the textual and acoustic information as intended (our evaluation in Section 5). Here, we can see textual features (red, Stage 1) are contaminated by acoustic features when we exclude information loss in training, which confirms the central role of the information loss term in our framework.
>
> __[If one needs to be confident that models do not use non-textual information while taking decisions, they can train models to make those decisions using pure transcripts]__
>
> Using pure transcription is not actually our goal. The accuracy of speech models is significantly better than that of text models for both emotion and speaker identification tasks, as both benefit much from acoustic features, and this is true in many other applications. Our goal is not to stop the model from using text, but rather to separate acoustic and textual features for controllability and interpretability; for example, we can see if a response of the dialog system is dependent on acoustic features or even reveal which layers and frames contribute textually or acoustically (as shown in the submission).

---

> > ### Author Response · Authors · 2024-11-25
> > **Response to reviewer HyNY (2)**
> >
> > __Q1: [Is there any evidence that Whisper is likely to have as entangled representations as SSL models]__
> >
> > While the Whisper model is trained in a supervised way for transcription tasks, the encoder part of Whisper is still expected to encode acoustic features as they only receive audio as input. Previous work has also shown that the Whisper encoder is capable of encoding both acoustic and linguistic features [1,2]. So the speech representations are still entangled.
> >
> > [1]: https://www.isca-archive.org/interspeech_2023/yang23d_interspeech.pdf  [2]: https://aclanthology.org/2023.emnlp-main.513.pdf
> >
> > __Q2: [Using standard some dev/test-{clean,other} splits]__
> >
> > The full Librispeech dev/test-{clean, other} splits are extensive, while for our purposes, a few hours of data are sufficient to train a classifier for decoding transcription. For test, however, we updated Table 1 with reporting WER on the entire LibriSpeech-test-clean and we reported the performance for the test set of Common Voice in Table B.1 in the appendix.
> >
> > __Q3: [Having a single linear probing classifier gets WER of ~50 for W2V2 and HuBERT. Only the pre-finetuned models get reasonable error rates. Is this a good evaluation setup to draw conclusions from?]__
> >
> > Here, we use VIB to retain relevant information and separate them to the extent that they are already present in the original representations, rather than encoding additional, extraneous features. Consequently, for pre-trained models, textual information will be separated from acoustic features to the degree that they exist in the original representations. Furthermore, including pre-trained models in the experiments enables a meaningful comparison with fine-tuned models, highlighting the extent to which textual and acoustic information contribute to the target task both before and after fine-tuning.
> >
> > __Q4: [What dataset is used to calculate WER in Table 1? Is this a mix of LibriSpeech and CommonVoice? Those are very different datasets, it would make sense to report them separately.]__
> >
> > We updated Table 1 with reporting WER on the entire LibriSpeech-test-clean and we reported the performance for the test set of Common Voice separately in Table B.1 in the appendix.
> >
> > __Q5: [The transcription vs gender/emotion classification task split is not a particularly convincing combination & Can replace it with an (ASR + text classifier) model]__
> >
> > Yes, if we're only interested in textual features, then training a text classifier on top of the generated transcription would work; however, in speech applications, we would like to incorporate acoustic features as well as speech content. But, perhaps the target task does not require all those acoustic features; in this case, those acoustic features can be suppressed to some extent (but not completely). For example, we can see in Figure 2 in the paper that not all the acoustic features encoded in the original speech representations are necessary for emotion recognition, so they were removed in stage 2.

---

> > > ### Author Response · Authors · 2024-11-29
> > >
> > > We would like to thank you again for your detailed and thoughtful feedback. We believe we have addressed the concerns you and the other reviewers raised, and we kindly ask that you review the improvements and clarifications we have made.
> > > Specifically, we have:
> > > - Clarified our motivation and the Targeted Setting / Desiderata.
> > > - Strengthened the experimental evaluation by incorporating comparison to previous methods (FACodec and SpeechSplit) in Section F in the appendix (Figures F.1 and F.2).
> > > - Thoroughly revised the Related Work discussion to provide a clearer explanation of how our approach aligns with, diverges from, and contributes to the existing body of research on disentanglement in speech processing.
> > >
> > > Please let us know if our responses properly address your concerns.

---

### Author Response · Authors · 2024-11-25
**General response to all reviewers**

We sincerely appreciate the detailed and thoughtful feedback from the reviewers, which has been instrumental in improving our paper.

Several reviews raised questions regarding the motivation for our setting and its relation to previous approaches for introducing disentanglement in speech modeling. Although we stand by our results and the relevance and novelty of our technical contribution, the reviews helped us realize that the framing of the paper and the discussion of and comparison to related work could be much improved. We have therefore made significant revisions to the paper (__marked in blue__), and hope all reviewers are willing to take the improvements and clarifications into account for their final assessments.

Specifically, we have:
- __Clarified the Targeted Setting / Desiderata:__ We have refined the description of the settings we are addressing, explicitly outlining the key desiderata of our proposed method.
- __Strengthened the Experimental Evaluation:__ We incorporated comparison to previous methods (FACodec and SpeechSplit) in Section F in the appendix (Figures F.1 and F.2). We show that, unlike the latent representations produced by our framework, the acoustic components of the previous work perform poorly in linear decoding of acoustic features, and their content vectors exhibit random performance in decoding audio transcriptions.
- __Enhanced the Related Work Discussion:__ The related work section has been thoroughly revised to provide a clearer explanation of how our approach aligns with, diverges from, and contributes to the existing body of research on disentanglement in speech processing.

We also provide responses to each individual reviewer.  We would appreciate it if you could let us know whether our responses properly address your concerns.

---

### Meta-Review · Area_Chair_ewVQ · 2024-12-18

**Metareview:**

This paper proposes to use VIB (Variational Information Bottleneck) as a framework for disentangling speech representations from neural speech models (like Wav2Vec2 and HuBERT) into two distinct components: textual content (what can be transcribed as text) and acoustic features (typically related to emotion or speaker identity). This separation could allow suppression of sensitive (speaker-related) information in future real-world applications.
Authors present a two-stage training framework. In the first stage, a decoder is trained to map speech representations to text while minimizing irrelevant information from input, ensuring only features necessary for transcription are preserved. In the second stage, another decoder is trained that has access to the textual representation from previous stage and is trained to predict target labels for downstream task while minimizing information encoding. Authors evaluated their framework on emotion recognition and speaker identification tasks, demonstrating that the resulting representations were effectively disentangled - the textual representations could predict transcriptions but performed randomly when predicting acoustic features, while acoustic representations showed the opposite pattern. Authors also analyzed how different layers of pre-trained and fine-tuned Wav2Vec2 models contribute to emotion recognition. They found that in models fine-tuned for automatic speech recognition (ASR), the acoustic contribution to emotion recognition decreases in higher layers while the textual contribution increases. Additionally, they showed that their framework can serve as a feature attribution method to identify the most significant frame representations for a given task, distinguishing between textual and acoustic contributions.

Strengths
- The proposed approach is well motivated (after revisions) and clearly described.
- There are a number of interesting analyses based on probing experiments to attempt to identify what information is still available in different layers of the network and assessment of the information related to distinct tasks in different frames of the input audio.

Weaknesses
- The multi-stage training procedure is cumbersome, since ordering matters
- Results seem somewhat inconsistent and require lots of post-hoc explanations. Additional validations would help.
- There is previous work using the Information Bottleneck for feature disentanglement

Authors were able to address many concerns in the rebuttal and reviewers appreciated these efforts. The main remaining weakness is that authors were not able to address all concerns in an updated revision. Given the wealth of results and comparisons, and the fact that authors want to make a claim about a very specific setting (with limited technical novelty due to VIB being a known technique), the AC does not recommend acceptance at this time, although reviews would not preclude it.

**Additional Comments On Reviewer Discussion:**

Authors provided several additional analyses and results during rebuttal period, mostly in response to reviewer requests (https://openreview.net/forum?id=xJc3PazBwS&noteId=HqaHeVkQ4p) Reviewers agreed with the authors choice of narrowly defining the scope of their work, but did not raise scores consistently and significantly in response.

---

### Decision · Program_Chairs · 2025-01-22

Reject